# Hippocampal Volume in Provisional Tic Disorder Predicts Tic Severity at 12-Month Follow-up

**DOI:** 10.3390/jcm9061715

**Published:** 2020-06-03

**Authors:** Soyoung Kim, Deanna J. Greene, Carolina Badke D’Andrea, Emily C. Bihun, Jonathan M. Koller, Bridget O’Reilly, Bradley L. Schlaggar, Kevin J. Black

**Affiliations:** 1Department of Psychiatry, Washington University School of Medicine, St. Louis, MO 63110, USA; kimsoyoung@wustl.edu (S.K.); dgreene@wustl.edu (D.J.G.); emilybihun@wustl.edu (E.C.B.); kollerj@wustl.edu (J.M.K.); bridget.oreilly@wustl.edu (B.O.); 2Department of Radiology, Washington University School of Medicine, St. Louis, MO 63110, USA; 3Kennedy Krieger Institute, Baltimore, MD 21205, USA; schlaggar@kennedykrieger.org; 4Department of Neurology, Johns Hopkins University School of Medicine, Baltimore, MD 21205, USA; 5Department of Pediatrics, Johns Hopkins University School of Medicine, Baltimore, MD 21205, USA; 6Department of Neurology, Washington University School of Medicine, St. Louis, MO 63110, USA; 7Department of Neuroscience, Washington University School of Medicine, St. Louis, MO 63110, USA

**Keywords:** Tourette syndrome, tic disorders, Provisional Tic Disorder, hippocampus, prognosis

## Abstract

Previous studies have investigated differences in the volumes of subcortical structures (e.g., caudate nucleus, putamen, thalamus, amygdala, and hippocampus) between individuals with and without Tourette syndrome (TS), as well as the relationships between these volumes and tic symptom severity. These volumes may also predict clinical outcome in Provisional Tic Disorder (PTD), but that hypothesis has never been tested. This study aimed to examine whether the volumes of subcortical structures measured shortly after tic onset can predict tic symptom severity at one-year post-tic onset, when TS can first be diagnosed. We obtained T1-weighted structural MRI scans from 41 children with PTD (25 with prospective motion correction (vNavs)) whose tics had begun less than 9 months (mean 4.04 months) prior to the first study visit (baseline). We re-examined them at the 12-month anniversary of their first tic (follow-up), assessing tic severity using the Yale Global Tic Severity Scale. We quantified the volumes of subcortical structures using volBrain software. Baseline hippocampal volume was correlated with tic severity at the 12-month follow-up, with a larger hippocampus at baseline predicting worse tic severity at follow-up. The volumes of other subcortical structures did not significantly predict tic severity at follow-up. Hippocampal volume may be an important marker in predicting prognosis in Provisional Tic Disorder.

## 1. Introduction

Tic disorders are neurodevelopmental disorders defined by the presence of tics: sudden, rapid, recurrent, non-rhythmic movements or vocalizations [1]. Tics are very common, appearing in at least 20% of elementary school-aged children [2]. Provisional Tic Disorder (PTD) is diagnosed when tics have been present for less than one year. While most children experience improvement in tic symptoms within the first few months after tic onset, some children continue to have tics for more than one year, meeting criteria for a Persistent (Chronic) Tic Disorder or Tourette’s Disorder (hereafter referred to as Tourette syndrome, “TS”). For children with persisting tics, severity can be quite variable across individuals, with some experiencing a significant worsening of tic symptoms that can impair quality of life [3]. Better prognostic ability in PTD may lead to patient-specific treatment, with treatment focused on those who are at greater risk of an increase in tic symptoms. Identifying biomarkers of tics may be key in improving this prognostic ability. However, studies in search of tic biomarkers have primarily compared people with TS to a control sample, identifying significant differences in neurophysiological measures, such as brain anatomy or function. Findings from such studies cannot disentangle whether the differences are due to an underlying cause of tics or secondary, compensatory changes that occur with the prolonged presence of tics. By contrast, biomarkers identified at the onset of tic symptoms are more likely to be related to the primary cause of tics. Thus, the goal of the current study was to identify volumetric Magnetic Resonance Imaging (MRI) biomarkers that can predict one-year tic outcome in children with recent-onset tics (i.e., tic duration < 9 months). No such study has been performed in PTD.

However, a large body of research has used structural MRI to measure volumes of subcortical brain structures in TS, after tics have become chronic. These cross-sectional studies examined group differences in subcortical volumes between people with diagnosed TS compared to controls, and have generated conflicting results. One finding that has received substantial attention is reduced caudate nucleus volume in TS in both children [4,5] and adults [5,6,7]. However, a more recent, large, multi-site study found no significant differences in caudate volume between children with TS and age-matched controls [8]. Volumetric findings in other subcortical structures have been discrepant as well. Some studies reported smaller volumes in the putamen [5,9], thalamus [4,10], and hippocampus [11], while others reported larger volumes in the putamen [11,12,13], thalamus [8,14,15], hippocampus [16,17], and amygdala [16,18]. These discrepant results may be partially due to sample differences, including comorbid conditions, medication use, and length of time with tics.

Moreover, MRI is highly susceptible to motion artifact, a highly problematic methodological issue when studying children with a movement disorder. Most previous volumetric MRI studies in TS excluded images with visually obvious motion contamination. Yet residual motion artifact even after visual screening can lead to spurious results, such as smaller volume estimates in individuals with more head movement during the scan [19,20]. Thus, it is possible that discrepant results were influenced by motion artifact. Notably, our large, multi-site volumetric study of children with TS started with structural MRI scans from 230 children with TS, but excluded 121 of these children due to visible motion artifact in the image [8]. This restricted, yet still large, sample of 103 children with TS (an additional 6 children were excluded for lack of a matched control) showed no differences in caudate volume compared to 103 age-matched controls. Thus, advances in quality control may call certain findings into question. Future studies must implement continually developing methods to better account for motion, such as prospective motion correction scan sequences [21].

Even if findings were not discrepant, and differences in brain structure between TS and controls were conclusive, it is impossible to resolve whether differences identified with cross-sectional studies could serve as predictive biomarkers useful for prognosis or clinical care. Longitudinal designs are necessary, as well as studying children at the onset of tic symptoms. The only longitudinal volumetric MRI study of children with TS found that a smaller caudate nucleus in childhood predicted more severe tics and other symptoms an average of 7.5 years later [9]. However, this hypothesis has not been tested in children within the first year of tic onset.

We hypothesized a priori (https://osf.io/y5vxj) that a smaller caudate volume in children with recent-onset tics (hereafter “NewTics”) would predict worse tic outcome at the one-year anniversary of tic onset, i.e., that tics would worsen or show less improvement. We extended our investigation beyond this one a priori hypothesis and examined whether the volume of other subcortical structures could predict tic outcome in children with recent-onset tics. In order to reduce motion artifact, we adopted prospective motion correction (vNavs sequences [21]) in our most recent data collection, in addition to careful quality control of all scans (with and without vNavs sequences).

## 2. Methods

### 2.1. Participants

The NewTics project is a longitudinal study of recent-onset tics conducted at Washington University School of Medicine, St. Louis, Missouri, USA (www.newtics.org) [22]. Children with recent onset of tic disorder often do not seek immediate medical attention, so even with active community recruitment, it was necessary to enroll subjects over a period of years. Here, we report the results of structural MRI data collected between September, 2010, and December, 2019. We enrolled children aged 5–10 years in three different groups: (1) NewTics group: children with tic onset within 9 months of study participation; (2) TS group: children with tics for more than one year, i.e., meeting criteria for Tourette’s Disorder or Persistent Tic Disorder; (3) tic-free control group: children with no tics as assessed by parent and self-reported history, clinical examination, and audiovisual observation. To increase the sample size of the TS and tic-free control groups, we included 11 children with TS and 22 tic-free children who had previously participated in other studies at Washington University School of Medicine. Our starting sample included 54 participants in the NewTics group, 38 participants in the TS group, and 41 participants in the tic-free group. After scan quality control (see Scan QC below), 41 NewTics, 34 TS, and 40 tic-free participants remained for analyses. Table 1 shows the characteristics of these participants and Table 2 shows symptom status for the NewTics group at the baseline and 12-month follow-up visits. The study was approved by the Washington University Human Research Protection Office (IRB), protocol numbers 201109157 and 201707059. Each child assented and a parent (guardian) gave informed consent. Additional MRI scan data and corresponding clinical information were shared with informed consent from other studies, including CTS (IRB 201412136), MSCPI (IRB 201402100), TRACK (IRB 201301004, 201808060), or NEWT (IRB 201601135).

### 2.2. Procedure

This study consisted of baseline and 12-month follow-up study visits. The baseline visit consisted of neuropsychological tests and clinical examination on one day (the full list of assessments has been reported in our previous work [3]) and an MRI scan visit (functional and structural MRI) within one week of the baseline visit. Clinical examination was repeated at a follow-up session 12-months after the best estimate date of the first definite tic. All participants who completed the study by December 2019 were included in the current report. 

### 2.3. MRI Acquisition

To minimize scan-day discomfort and head movement, participants entered a mock scanner on the day of their clinical examination. On the MRI day, scans lasted approximately one hour to collect T1- and T2-weighted structural images, resting-state functional MRI, and pseudo-continuous arterial spin labeling (pCASL) images. Scan quality was checked immediately after the acquisition, and sequences were repeated if necessary. In the current study, high-resolution T1-weighted magnetization prepared rapid gradient echo (MPRAGE) images covering the whole brain were analyzed. Different scanners and sequences were used over the 9 years of data acquisition (Cohort 1: Siemens TRIO 3T MRI scanner, 176 slices, FOV = 224 × 256, 1 mm isotropic voxels, TR = 2200 ms, TE = 2.34 ms, TI = 1000 ms, and flip angle = 7 degrees; Cohort 2: Siemens Prisma 3T MRI scanner, 196 slices, FOV = 240 × 256, 0.8 mm isotropic voxels, TR = 2400 ms, TE = 2.22 ms, TI = 1000 ms, and flip angle = 8 degrees; Cohort 3: Siemens Prisma 3T scanner, 196 slices, FOV = 256 × 256, 1 mm isotropic voxels, TR = 2500 ms, TE = 2.9 ms, TI = 1070 ms, and flip angle = 8 degrees). Importantly, 25 NewTics, 27 TS, and 19 tic-free control participants were scanned with a prospective motion correction sequence (vNavs [23]; Cohort 3). We also included T1-weighted MPRAGE images from 11 children with TS and 22 children without tics (including 11 participants scanned with a vNavs sequence) from other studies. Detailed scan parameters are shown in Appendix B. If the participant had more than one T1 scan, the scan with the better QC rating was used for the analysis.

### 2.4. Scan QC

In order to assess scan quality, we extracted MRIQC (MRI Quality Control) [24] from each T1 scan. Among 64 image quality metrics of MRIQC, we found that the average signal-to-noise ratio (SNR) of gray matter, white matter, and cerebrospinal fluid (CSF) (hereafter “SNR (total)”) calculated using the within-tissue variance was highly correlated with subjective rating by visual inspection following standardized criteria [25]. We excluded T1 scans with scan rating C3 (fail) or SNR total below 7.5 from the further analysis (see Appendix C). Thus, 13 NewTics, 4TS, and 2 tic-free participants were excluded.

### 2.5. Analysis

We used a fully automatic segmentation tool, volBrain [26], which segments and quantifies the volumes of subcortical structures including the putamen, caudate, pallidum, thalamus, hippocampus, amygdala, and nucleus accumbens. volBrain showed superior accuracy in segmenting all seven subcortical structures [26] compared to other publicly available software packages, FreeSurfer [27] and FSL-FIRST [28]. Although the hippocampus is known to be difficult to segment [29], volBrain achieved high dice similarity indices in comparison to manual segmentation in segmenting the hippocampus [30].

VolBrain also estimates total intracranial volume (ICV). We adopted the residual approach [31] to control for interindividual head size differences (see Appendix D). As we did not hypothesize asymmetry to be of interest, we summed left and right hemisphere volumes for each structure. Total (left + right) regional volumes adjusted for ICV were the dependent variables. (An exploratory analysis of each left and right subcortical structure appears in Appendix E). We conducted multiple regression analyses within the NewTics participants to test whether subcortical structure volume at the baseline visit could predict tic severity at the follow-up visit. Baseline total tic score from the Yale Global Tic Severity Scale (YGTSS), age, sex, attention-deficit/hyperactivity disorder (ADHD) diagnosis, obsessive–compulsive disorder (OCD) diagnosis, and scanner were included as covariates, but insignificant terms were eliminated via backward stepwise regression. Group comparisons were conducted using one-way ANOVA. We also conducted independent t-tests specifically comparing NewTics vs. tic-free and TS vs. tic-free. As we did not correct for multiple comparisons, we added Bayesian hypothesis testing with the Bayesian information criterion (BIC) method. BF_10_ over 3 was considered as positive [32]/substantial [33] evidence (strong evidence if BF_10_ > 10) [34]. We used JASP (JASP Team. JASP Version 0.9, https://jasp-stats.org/) for Bayesian hypothesis testing and SPSS for all other statistical analyses.

## 3. Results

### 3.1. Mean Clinical Change

Consistent with our previous report on 20 overlapping participants [3], NewTics participants’ tic symptoms improved on average between the baseline and follow-up visits. The mean YGTSS total tic score decreased by approximately 22% (Table 2).

### 3.2. Predictors of Change in the NewTics Group

Baseline hippocampal volume, but no other structure volumes, significantly predicted total tic score at the 12-month follow-up visit, after controlling for the baseline tic symptoms (R² = 0.492, F(1,38) = 18.38, *p* < 0.001; adjusted R² = 0.465) (Figure 1, first row). The estimated Bayes factor BF_10_ was 16.88, indicating strong evidence in favor of adding hippocampal volume to the null model with baseline tic symptoms alone. Stepwise regression analysis was conducted to test whether age, sex, handedness, comorbid ADHD diagnosis, OCD diagnosis or scanner significantly improved the model, but none of these factors were selected. The final model is shown in Table 3. This result was not due to an association already present at baseline. Cross-sectional analyses to examine the relationship between the volumes of subcortical structures and the total tic score within the baseline session revealed no significant association in any subcortical structure volumes (*p* ≥ 0.25; Figure 1, second row). An exploratory analysis examining hippocampal subfields using the volBrain HIPS pipeline [35] showed similar results for baseline CA1 volume and for combined CA2 + CA3 volume (Appendix F). A similar post-hoc analysis found that either left or right hippocampal volume at baseline significantly predicted follow-up total tic score (Appendix E).

### 3.3. Group Comparisons

Figure 1 (bottom row) shows putamen, caudate, nucleus accumbens, globus pallidus, amygdala, thalamus, and hippocampus volumes for each group. One-way ANOVAs revealed no significant main effects of the group in any subcortical structure (*p* ≥ 0.113). As we specifically hypothesized that children with tics (NewTics group and TS group) would differ from tic-free children (H1), we compared the NewTics and TS groups to the tic-free group separately using independent t-tests. Hippocampal volume differed between the NewTics and tic-free control groups (*t*(79) = 2.022, *p* = 0.047). The estimated Bayes factor BF_10_ was 1.40, indicating weak evidence in favor of the alternative hypothesis (H1). There was no significant difference between NewTics and tic-free in other subcortical structures (minimum *p* = 0.122) or between TS and tic-free participants in any subcortical structures (minimum *p* = 0.116).

### 3.4. Subgroup Analysis

We conducted a subgroup analysis with the participants whose T1 scans were collected with prospective motion correction (vNavs) sequences. We included the participants from our own NewTics study only, as we carefully screened our tic-free controls for tics using structured interviews, direct examination, and video monitoring of the child sitting alone. The main results with hippocampal volume were still present. The analysis within the NewTics group showed that hippocampal volume predicted the total tic score at the 12-month follow-up visit, after controlling for the baseline tic symptoms (R² = 0.618, F(1,22) = 17.8, *p* < 0.001; adjusted R² = 0.583; see Figure 2, lower figures). We found no significant association between the hippocampal volume and the total tic score within the baseline session (*r* = 0.128; *p* = 0.542; see Figure 2, lower figures). One-way ANOVA revealed a significant main effect of group for hippocampal volume (*p* = 0.018); see Appendix G); post-hoc tests showed greater hippocampal volume in each patient group compared to controls (NewTics vs. tic-free: *t*(42) = 2.66, *p* = 0.011; TS vs. tic-free: *t*(40) = 2.23, *p* = 0.027, uncorrected). The subgroup analyses for other subcortical structures are shown in Appendix G.

## 4. Discussion

The goal of the current study was to investigate whether the volume of subcortical structures in children with recent-onset tics predicted tic outcome at the one-year anniversary of tic onset, when Tourette’s Disorder or Persistent Tic Disorder can first be diagnosed. The study design and a summary of the results are shown in Figure 2. We found that hippocampal volume measured within months of tic onset predicted one-year tic outcome, such that children with a larger hippocampus showed worse tic outcome (less improvement). Volumes of other subcortical structures did not significantly predict tic outcome. We also examined whether the volumes of any subcortical structures differed between NewTics, TS, and tic-free groups. While hippocampal volume differed between NewTics and controls, it was near the threshold of statistical significance. No significant difference was found in any other subcortical structures.

Our a priori hypothesis regarding caudate volume was not supported. Smaller caudate volumes in children and adults with TS have been repeatedly reported (reviewed in Greene et al. [36]), and a longitudinal study showed that smaller caudate volume in children with TS predicted worse tic outcome in young adulthood [9]. The different patterns of prognosis might be due to the different phase of illness, or different periods of follow-up. While we studied the prognosis of children presenting within a few months of tic onset, measured at one year, Bloch et al. [9] examined subjects at least a year after tic onset, with a follow-up mean of 7.5 years later.

MR images acquired during even small head motion can lead to artifactually smaller volumes [19,20], raising concerns about studies that did not specify how carefully they controlled scan quality. We adopted a prospective motion correction sequence (vNavs [23]) to reduce the impact of head motion, and also excluded the scan images with low SNR from the analysis. Within this carefully controlled dataset, we found no significant group difference in caudate volume or its association with tic symptoms. Thus, previous findings of smaller caudate volume might be partially due to the individuals with tics moving more inside the MRI scanner. Alternatively, the lack of significance in the current study may reflect type II error, but one of the two largest studies similarly found no significant reduction in caudate volume in children with TS [8].

In the current study, the significant association between baseline volumes and tic symptom severity at follow-up was specific to the hippocampus even when comorbidities were statistically controlled. Analysis of left and right hemisphere separately revealed the same pattern of results. Hippocampal enlargement in children with TS has been reported previously [16]. Hippocampal volume quantified at the baseline visit was not associated with the tic symptom severity at the baseline visit. Rather, volume was correlated with tic symptom severity at the 12-month visit, suggesting that hippocampal volume may be related to the persistence of tic symptoms, but not the initial acquisition of tics. This finding is consistent with the idea that tics are thought to result from aberrant habit learning [37]. Both tics and habits are inflexible, repetitive behaviors that are acquired over a period of time. Given these similarities, a behavioral study using a motor learning and memory task reported a negative correlation between the rate of forgetting (unlearning) and motor tic severity [38]. Children and adolescents with severe tics showed evidence of enhanced motor memory, in that they took longer to unlearn previously learned motor patterns of behavior. The hippocampus plays a role in memory consolidation not just in the cognitive domain but also in the motor domain [39]. Together with the previous behavioral finding, our results suggest that tics, once they develop, may be more likely to persist in children with a larger hippocampus.

The apparent lack of a significant group difference between the TS and tic-free groups is complicated. If the hippocampus is related to the main cause of tic symptom persistence, then greater hippocampus volume would be expected in the TS group compared to tic-free controls. This lack of significant group difference may indicate that the hippocampus plays a critical role in initial tic symptom persistence up to approximately a year after tic onset, but thereafter the relationship between the hippocampal volumes and tic symptoms may be more complex. For example, ADHD, OCD [40,41,42], and anxiety disorder [43]—all of which frequently co-occur with tic disorders—have been associated with reduced hippocampal volume. However, in the current study, these clinical subgroups (among subjects whose comorbid symptom records were available) did not differ in terms of hippocampal volume. Comorbidity and age may also affect the relationship between hippocampal volume and tics. Although Peterson et al. found larger hippocampal volume in children with TS, some subregions became smaller compared to controls by adulthood [16]. Further, reduced hippocampal volumes in TS have been reported in adolescents [44] and in adults with comorbid OCD [17]. On the other hand, the subset of participants whose data were collected using the prospective motion correction MR sequence, and with tics carefully screened by a semi-standardized diagnostic interview (K-SADS), direct examination, and video recording of the child sitting alone, revealed increased hippocampal volumes in the NewTics and TS groups compared to the tic-free group (Appendix A). Further studies are needed to determine whether additional data collected with this improved methodology confirm this potential group difference. Prospective motion correction is advantageous because it can acquire the scan data with adequate quality even in those participants with some head motion, while scan quality control after acquisition may bias the sample by excluding the participants with more severe tic symptoms.

A recent genetics study by Mufford et al. examined the genetic variation that increases risks for TS and its influence on the volume of specific subcortical brain structures [45]. One of their analyses revealed evidence of pleiotropy between TS risk variants and variants associated with greater hippocampal volume. This result is in line with our finding that a larger volume of the hippocampus was associated with worse tic outcomes.

One limitation of the current study is that we collected MRI scans using different scanning sequences over 9 years of recruiting. A retrospective study estimated the typical delay from the onset of tics to diagnosis as 10 years [46]. This result highlights the difficulty in recruiting the PTD population. The NewTics group’s total sample of 54 participants includes 29 participants who were recruited in the first 6 years while MRI methods improved. Since receiving R01 funding for the NewTics study, allowing more vigorous recruiting including local advertisement, and obtaining access to the vNavs prospective motion correction method, we have used identical scan parameters for the most recent 25 participants. Importantly, all of the 25 participants with vNavs scans passed scan QC while 13 of the first 29 participants failed scan QC. This suggests that the vNavs sample may include less selection bias for tic symptom severity, as the vNavs sequence allows collection of adequate quality MRI scans even from participants with greater head motion. To prevent a possible influence of varying scan parameters on the results, we conducted a subgroup analysis limited to the most recent 25 participants, with the vNavs sequence, and found the same prognostic value of hippocampal volume. The vNavs scan subgroup analysis comparing the NewTics group to tic-free controls also revealed a significant group difference in hippocampal volume that was not evident in the entire sample analysis.

In summary, our results suggest that hippocampus volume may be a critical biomarker predicting tic symptom persistence in children with Provisional Tic Disorder. Further studies with longer follow-up are required to understand more completely the longitudinal relationship between hippocampal volume and tic symptoms.

## Figures and Tables

**Figure 1 jcm-09-01715-f001:**
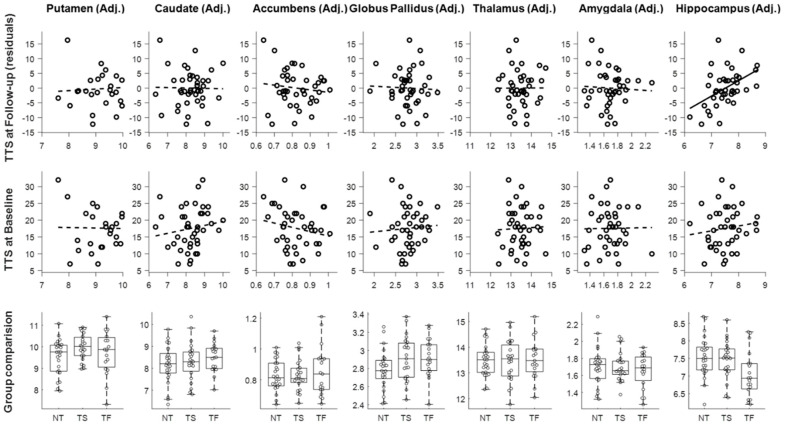
Volumes of subcortical structures adjusted for intracranial volume (ICV) and prognosis analysis (top row), cross-sectional analysis (middle row), and group comparison analysis (bottom row). TTS: total tic score; NT: NewTics; TS: Tourette syndrome; TF: Tic-free.

**Figure 2 jcm-09-01715-f002:**
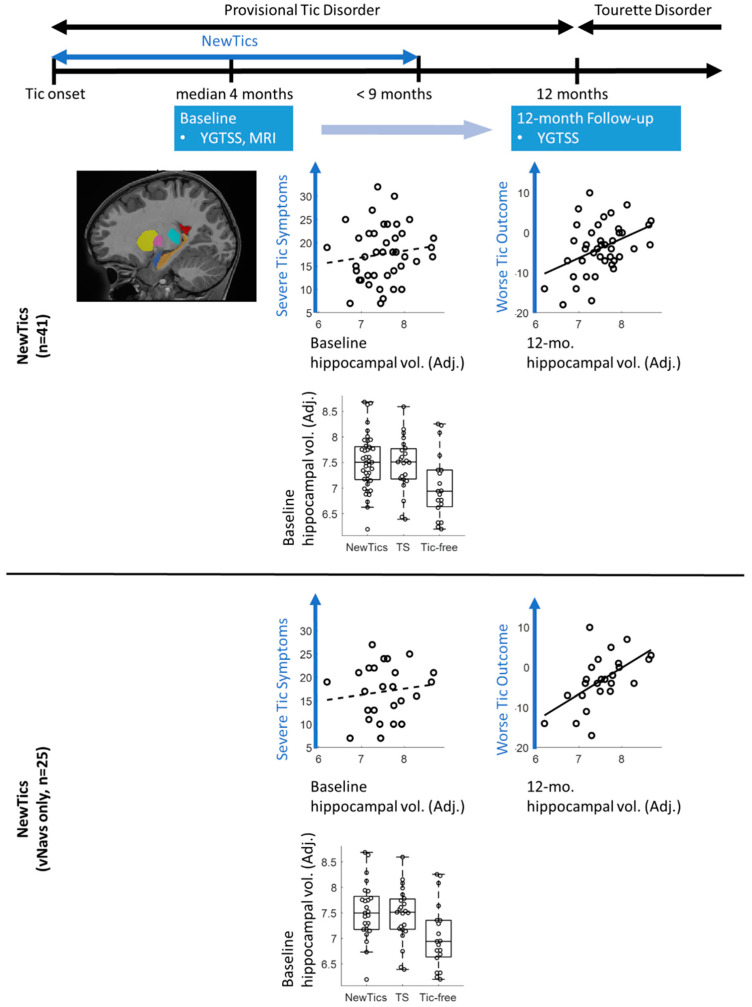
A graphical representation of the study design and a summary of the results. The upper figures show the results of the entire set of participants (*n* = 41). The lower figures represent the subgroup analysis consisting of the scans collected with vNavs (*n* = 25). YGTSS: Yale Global Tic Severity Scale; MRI: Magnetic Resonance Imaging; TS: Tourette syndrome.

**Table 1 jcm-09-01715-t001:** Characteristics of the participants in the NewTics, TS, and tic-free groups.

Variable	NewTics	TS	Tic-Free
*N*	41	34	40
Sex	30 M/11 F	25 M/9 F	29 M/11 F
Age	7.87 ± 1.61 (5.41–10.81)	8.33 ± 1.55 (5.11–10.99)	8.18 ± 1.51 (5.19–10.92)
Tic duration (year)	0.34 ± 0.16 (0.07–0.73)	3.16 ± 1.67 (1.07–6.63) (*n* = 23) *	*n*/a
YGTSS total tic score (TTS)	17.59 ± 6.10 (7–32)	18.59 ± 6.54 (7–30)	*n*/a
YGTSS impairment	8.29 ± 8.56 (0–30)	11.17 ± 12.78 (0–40) (*n* = 30) *	*n*/a
ADHD diagnosis	14	17 of 30 *	10 of 26 *
OCD diagnosis	3	5 of 30 *	0 of 26 *
*N* with brain active medications	9	10 of 30 *	8 of 26 *

* Full clinical data were not available for some participants whose data came from other studies. TS: Tourette syndrome; YGTSS: Yale Global Tic Severity Scale; ADHD: Attention Deficit Hyperactivity Disorder; OCD: Obsessive-compulsive disorder; M: male; F: female.

**Table 2 jcm-09-01715-t002:** Characteristics of the NewTics group participants at the baseline and 12-month follow-up session.

Variable	Baseline Visit	12-Month Follow-Up
*N*	41	41
Tic duration (days)	123.07 ± 58.52 (25–268)	371.71 ± 11.13 (355–409)
YGTSS total tic score (TTS)	17.59 ± 6.10 (7–32)	13.78 ± 7.60 (0–37)
YGTSS impairment	8.29 ± 8.56 (0–30)	4.63 ± 6.84 (0–20)
DCI	33.24 ± 14.36 (12–80)	43.41 ± 15.85 (13–79)
PUTS	13.66 ± 5.39 (9–31) (*n* = 38) *	15.32 ± 5.65 (9–30)
ADHD rating scale (ARS)	13.41 ± 11.81 (0–40)	15.05 ± 11.92 (0–41)
ADHD diagnosis	14	17
CY-BOCS	3.95 ± 6.45 (0–26)	6.93 ± 8.62 (0–26)
OCD diagnosis	3	9
SRS	48.83 ± 10.01 (35–78)	*n*/a

* PUTS scores were not obtained from 3 young children at baseline visit because of difficulty in reporting these internal phenomena. DCI: Diagnostic Confidence Index; PUTS: Premonitory Urge for Tics Scale; CY-BOCS: Child Yale-Brown Obsessive Compulsive Scale; SRS: Social Responsiveness Scale.

**Table 3 jcm-09-01715-t003:** Stepwise regression analysis for prediction of tic severity at 12-month visit based on hippocampal volume at baseline visit and other baseline clinical variables.

Variable		B	SE_B_	β	*p*
*Y* = Total Tic Score at 12-Month Follow-up					
	Hippocampus volume (Adjusted)	5.311	1.627	0.381	0.002
	Total tic score at baseline session	0.676	0.145	0.542	<0.001
	Intercept	−38.02	12.197		0.003

B: unstandardized coefficients; SE: standard error; β: standardized coefficients.

## Data Availability

The supplementary data file provides individual participant data. Data used in the preparation of this article are from the NIH-supported NIMH Data Repositories in [DOI 10.15154/1518691], and nih.figshare.com (project 68282).

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
