# Peer review of "Hippocampal Volume in Provisional Tic Disorder Predicts Tic Severity at 12-Month Follow-up"

_jcm, 2020, doi:10.3390/jcm9061715_

Round 1

Reviewer 1 Report

This is a very interesting and unique study aiming to “predict” the course of tic severity in kids with provisional tic disorder. This is a difficult study to execute. I am not surprised it took several years. The authors are to be congratulated for their efforts.

The strengths of the work are numerous. In particular I liked the overall design, the careful QC procedures, the attention to motion artifacts and that the results replicated in the subsample using a prospective motion correction sequence.

I have one main conceptual issue and several other topics to discuss.

In my view, the use of the term prediction is problematic in the absence of a control group. This kind of causal language should be avoided. Rather, the most appropriate language would be correlation of x brain structure volume at baseline with observed changes in tic severity over time (or similar language).

The choice to sum the volumes of right and left subcortical structures is understandable but may also be problematic because the approach assumes that there are no significant asymmetries in these structures in TS. These have been demonstrated at least in other neuropsychiatric disorders such as autism or OCD (see ENIGMA papers). Could the authors consider additional exploratory analyses for left and right hemispheres?

Throughout the paper, it is important to clearly indicate which procedures/analyses/decisions were planned vs post-hoc. This can be simply achieved by adding “post-hoc” to the relevant sentences. For example, some of the analyses relating to hippocampal subfields fall under this category.

Relating to the above, is the spatial resolution good enough to reliably study hippocampal subfields? It is unclear what the meaning of the results is anyway. Or at least the authors do not discuss these results. Consider removing these analyses.

Line 228. Something wrong here: see Error! Reference source not found

I miss a section on strengths and limitations of the study. The use of different scanning sequences comes to mind as a potentially important limitation.

Reviewer 2 Report

The idea for this article is interesting. 

Tics are common in children, with a generally accepted prevalence rate of 20%, when tics have not yet lasted a year since onset provisional tic disorder is diagnosed (PTD). Some children with PTD  experience marked improvement or remission of tics within  the first year, while others go on to a diagnosis of Tourette Syndrome or chronic tic disorders. Children with recent onset tics and their parents want to know prognosis, however to my knowdlege very little research has been performed in this interesting field.

The paper is well written and comes from a well known group of expert in tic disorders,  but   the manuscript could be improved.

Sample size is small, data collection methods are rigorous and using the best measures available. However some of the analyses made the article difficult to follow, which is unfortunate because I think the topic is good.

Also recently Psychiatric genomic consortium_Tourette syndrome working group undertook a genome-wide investigation of the overlap between TS genetic risk and genetic influences on the volume of specific subcortical brain structures that have been implicated in TS ( Mufford M et al Traslational Psychiatry 2019). In particular the AA found three SNPs  variants associated with TS after conditioning on hippocampal (rs1922786) and ICV (rs2708146 and rs72853320) volumes  that have not previously been associated with a neuropsychiatric phenotype.   Authors should re write the discussion on the light of these results, because it's a big limitation that could interfere with all the paper.

Round 2

Reviewer 1 Report

Thank you for the helpful revisions. I find the results of the R and L hemispheres compelling. They add credibility to the findings.

For consistency with the way the hippocampal subfields results are presented, the laterality results should be briefly mentioned in the results section, even if the results are presented in detail in the appendix. Specifically, the text highlighted in appendix D could easily move to the results section. 

Similarly, I would recommend adding a brief sentence in the discussion that the results were significant for both R and L hippocampus.

Author Response

Thank you. We edited our results/discussion as suggested.

Reviewer 2 Report

The manuscript is improved and I suggest to reconsider it for the pubblication

Author Response

Thank you.